# IfQA: A Dataset for Open-domain Question Answering under Counterfactual Presuppositions

**Wenhao Yu**[♦*]**, Meng Jiang**[♣]**, Peter Clark**[♠]**, Ashish Sabharwal**[♠]
[♦]Tecent AI Seattle Lab; [♣]University of Notre Dame; [♠]Allen Institute for AI
[♦]wenhaowyu@global.tencent.com; [♠]ashishs@allenai.org

## Abstract

Although counterfactual reasoning is a fundamental aspect of intelligence, the lack of large-scale counterfactual open-domain question-answering (QA) benchmarks makes it difficult to evaluate and improve models on this ability. To address this void, we introduce the first such dataset, named IfQA, where each question is based on a counterfactual presupposition via an "if" clause. Such questions require models to go beyond retrieving direct factual knowledge from the Web: they must identify the right information to retrieve and reason about an imagined situation that may even go against the facts built into their parameters. The IfQA dataset contains 3,800 questions that were annotated by crowdworkers on relevant Wikipedia passages. Empirical analysis reveals that the IfQA dataset is highly challenging for existing open-domain QA methods, including supervised retrieve-then-read pipeline methods (F1 score 44.5), as well as recent few-shot approaches such as chain-of-thought prompting with ChatGPT (F1 score 57.2). We hope the unique challenges posed by IfQA will push open-domain QA research on both retrieval and reasoning fronts, while also helping endow counterfactual reasoning abilities to today's language understanding models. The IfQA dataset can be found and downloaded at https://allenai.org/data/ifqa.

## 1 Introduction

Counterfactual reasoning captures human tendency to create possible alternatives to past events and imagine the consequences of something that is contrary to what actually happened or is factually true (Hoch, 1985). Take, for example, the business arena where a corporate leadership team might rigorously analyze the potential ripple effects had they opted for an alternative investment strategy (Baron,

---

* The majority of the work was completed during Wenhao's internship at the Allen Institute for AI.

2000; Atherton, 2005). Counterfactual reasoning has long been considered a necessary part of a complete system for AI. However, few NLP resources aim at gauging the effectiveness of such reasoning capabilities in AI models, especially for the open-domain QA task. Instead, existing formulations of open-domain QA tasks mainly focus on questions whose answer can be deduced directly from global, factual knowledge (e.g., What was the occupation of Lovely Rita according to the song by the Beatles?) available on the Internet (Joshi et al., 2017; Kwiatkowski et al., 2019; Yang et al., 2018).

Counterfactual presupposition in open-domain QA can be viewed as a causal intervention. Such intervention entails altering the outcome of events based on the given presuppositions, while obeying the human readers' shared background knowledge of how the world works. To answer such questions, models must go beyond retrieving direct factual knowledge from the Web. They must identify the right information to retrieve and reason about an imagined situation that may even go against the facts built into their parameters.

Although some recent work has attempted to answer questions based on counterfactual evidence in the reading comprehension setting (Neeman et al., 2022), or identified and corrected a false presupposition in a given question (Min et al., 2022), none of existing works have been developed for evaluating and improving counterfactual reasoning capabilities in open-domain QA scenarios. To fill this gap, we present a novel benchmark dataset, named IfQA, where each of over 3,800 questions is based on a counterfactual presupposition defined via an "if" clause. Two examples are given in Figure 1. IfQA combines causal inference questions with factual text sources that are comprehensible to a layman without an understanding of formal causation. It also allows us to evaluate the capabilities and limitations of recent advances in QA methods in the context of counterfactual reasoning.

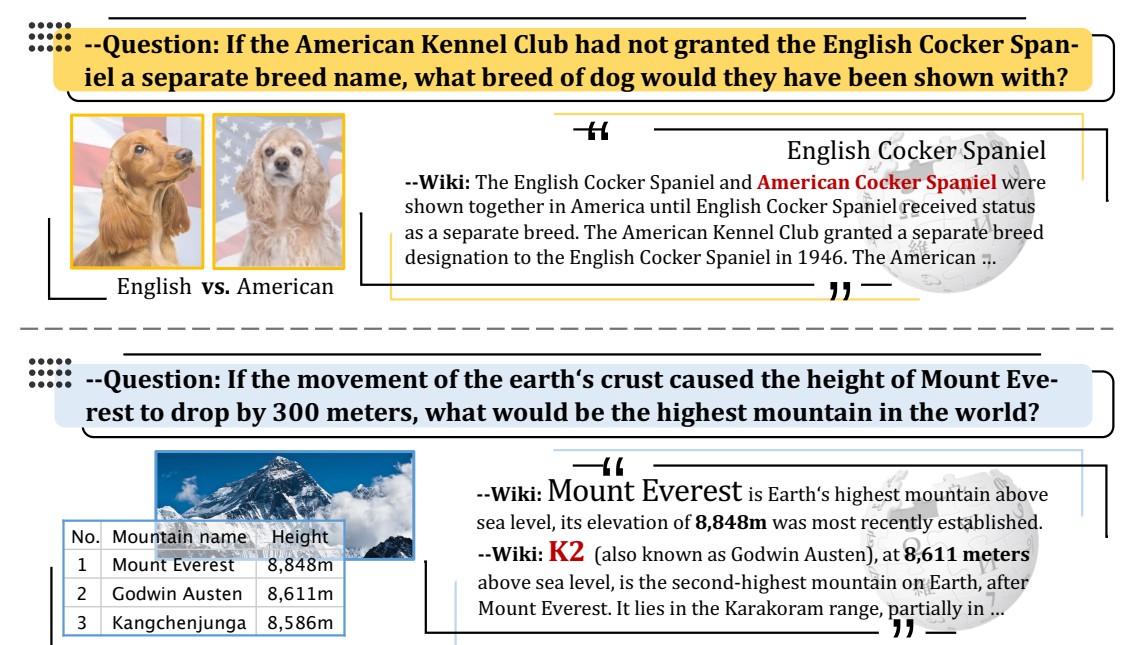

Figure 1: In the IfQA dataset, each question is based on a counterfactual presupposition via an "if" clause. To answer the question, one needs to retrieve relevant facts from Wikipedia and perform counterfactual reasoning.

IfQA introduces new challenges in both retrieval and reading. For example, to answer the 2nd example question in Figure 1, "If the movement of the earth's crust caused the height of Mount Everest to drop by 300 meters, which mountain would be the highest mountain in the world?", the search and reasoning process can be divided into four steps: (i) retrieve documents relevant to the current height of Mount Everest *(8,848 metres)*; (ii) calculate the height based the counterfactual presupposition *(8,848-300=8,548 metres)*; (iii) retrieve documents relevant to the current second-highest mountain in the world *(K2: 8,611 metres)*; and (iv) compare the heights of lowered Mount Everest and K2, then generate the answer *(K2)*.

To establish an initial performance level on IfQA, we evaluate both state-of-the-art close-book and open-book models. Close-book models, such as chain-of-thought (CoT) reasoning with Chat-GPT (Wei et al., 2022), generate answers and optionally intermediate reasoning steps, without access to external evidence. On the other hand, open-book models, such as RAG (Lewis et al., 2020) and FiD (Izacard and Grave, 2021), first leverage a retriever over a large evidence corpus (e.g. Wikipedia) to fetch a set of relevant documents, then use a reader to peruse the retrieved documents and predict an answer.

Our experiments demonstrate that IfQA is a challenging dataset for both retrieval as well as reading

and reasoning. Specifically, we make the following observations. First, in retrieval, traditional dense retrieval methods based on semantic matching cannot well capture the discrepancy between counterfactual presuppositions and factual evidence, resulting failing to retrieve the gold passages in nearly 35% of the examples. Second, state-of-the-art reader models, such as FiD, achieve an F1 score of only 50% even when the gold passage is contained in the set of retrieved passages. Third, close-book CoT reasoning can effectively improve the end-QA performance, but still heavily lags behind open-book models. Lastly, combining passage retrieval and large model reasoner achieves the best results.

We hope the new challenges posed by IfQA will help push open-domain QA research in an interesting new direction, as well as towards more effective general-purpose retrieval and reasoning methods.

## 2 Related Work

### 2.1 Open-domain Question Answering

The task of answering questions using a large collection of documents (e.g., Wikipedia) of diversified topics, has been a longstanding problem in NLP, information retrieval (IR), and related fields (Chen et al., 2017; Brill et al., 2002; Yu et al., 2022). A large number of **QA benchmarks** have been released in this space, spanning the different types of challenges represented behind them, in-

cluding single-hop questions (Joshi et al., 2017; Kwiatkowski et al., 2019; Berant et al., 2013), multi-hop questions (Yang et al., 2018; Trivedi et al., 2022), ambiguous questions (Min et al., 2020), multi-answer questions (Rubin et al., 2022; Li et al., 2022), multi-modal questions (Chen et al., 2020; Zhu et al., 2021a), real time questions (Chen et al., 2021; Kasai et al., 2022), and etc.

*To the best of our knowledge,* all existing formulations assume that each question is based on factual presuppositions of global knowledge. In contrast, the questions in our IfQA dataset are given counterfactual presuppositions for each question, so the model needs to reason and produce answers based on the given presuppositions combined with the retrieved factual knowledge. This makes IfQA a novel and qualitatively different dataset.

Mainstream open-domain **QA methods** employ a retriever-reader architecture, and recent follow-up work has mainly focused on improving the retriever or the reader (Chen and Yih, 2020; Zhu et al., 2021b; Ju et al., 2022). For the retriever traditional methods such as TF-IDF and BM25 explore sparse retrieval strategies by matching the overlapping contents between questions and passages (Chen et al., 2017; Yang et al., 2019). DPR (Karpukhin et al., 2020) revolutionized the field by utilizing dense contextualized vectors for passage indexing. Furthermore, other research improved the performance by better training strategies (Qu et al., 2021; Asai et al., 2022), passage re-ranking (Mao et al., 2021) and etc. Recent work has found that large language models have strong factual memory capabilities, and can directly generate supporting evidence in some scenarios, thereby replacing retrievers (Yu et al., 2023). Whereas for the reader, extractive readers aimed to locate a span of words in the retrieved passages as answer (Karpukhin et al., 2020; Iyer et al., 2021; Guu et al., 2020). On the other hand, FiD and RAG, current state-of-the-art readers, leveraged encoder-decoder models such as T5 to generate answers (Lewis et al., 2020; Izacard and Grave, 2021; Izacard et al., 2022).

## 2.2 Counterfactual Thinking and Causality

Causal inference involves a question about a counterfactual world created by taking an intervention, which have recently attracted interest in various fields of machine learning (Niu et al., 2021), including natural language processing (Feder et al., 2022). Recent work shows that incorporating coun-

terfactual samples into model training improves the generalization ability (Kaushik et al., 2019), inspiring a line of research to explore incorporating counterfactual samples into different learning paradigms such as adversarial training (Zhu et al., 2020) and contrastive learning (Liang et al., 2020). These work lie in the orthogonal direction of incorporating counterfactual presuppositions into a model's decision-making process.

In the field of NLP, existing counterfactual inferences are ubiquitous in many common inference scenarios, such as counterfactual story generation (Qin et al., 2019), procedural text generation (Tandon et al., 2019). For example, in TIME-TRAVEL, given an original story and an intervening counterfactual event, the task is to minimally revise the story to make it compatible with the given counterfactual event (Qin et al., 2019). In WIQA, given a procedural text and some perturbations to steps mentioned in the procedural, the task is to predict whether the effects of perturbations to the process can be predicted (Tandon et al., 2019). However, to the best of our knowledge, none of existing benchmark datasets was built for the open-domain QA.

## 3 IfQA: Task and Dataset

### 3.1 Dataset Collection

All questions and answers in our IfQA dataset were collected on the Amazon Mechanical Turk (AMT)[1], a crowdsourcing marketplace for individuals to outsource their jobs to a distributed workforce who can perform these tasks. We offered all AMT workers $0.8 per annotation task, which leads to $15 to $20 per hour in total. To maintain the diversity of labeled questions, we set a limit of 30 questions per worker. In the end, the dataset was annotated by a total of 188 different crowdworkers.

Our annotation protocol consists of three phases. First, we automatically extract passages from Wikipedia which are expected to be amenable to counterfactual questions. Second, we crowdsource question-answer pairs on these passages, eliciting questions which require counterfactual reasoning. Finally, we validate the correctness and quality of annotated questions by one or two additional workers. These phases are described below in detail, and the annotation task form is shown in Figure 3.

---

[1] https://www.mturk.com

Table 1: Example questions from the IfQA dataset, with the proportions with different types of answers.

| Answer Type | Passage (some parts shortened) | Question | Answer |
|---|---|---|---|
| Entity (49.7%) | LeBron James: ... On June 29, 2018, James opted out of his contract with the Cavaliers and became an unrestricted free agent. On July 1, his management company, Klutch Sports, announced that he would sign with the Los Angeles Lakers. | If LeBron James had not been traded to the Los Angeles Lakers, which team would he have played for in 2018-2019 season? | (Cleveland) Cavaliers |
| Number (15.9%) | 7-Eleven: ... Japan Co., Ltd. in 2005, and is now held by Chiyoda, Tokyo-based Seven & i Holdings. 7-Eleven operates, franchises, and licenses 71,100 stores in 17 countries as of July 2020. | If 7-Eleven expanded its reach to five more countries in 2020, how many countries would have 7-Eleven by the end of the year? | 22 (countries) |
| Date (14.5%) | 2020 Summer Olympics: ... originally scheduled to take place from 24 July to 9 August 2020, the event was postponed to 2021 in March 2020 as a result of the COVID-19 pandemic, ... | If Covid-19 hadn't spread rapidly across the globe, when would the Tokyo Olympics in Japan start? | July 24, 2020 |
| Others (19.9%) | 1991 Belgian Grand Prix: Patrese's misfortune promoted Prost to second, with Nigel Mansell third, Gerhard Berger fourth, Alesi fifth, and Nelson Piquet sixth while the sensation of qualifying, Schumacher, was an amazing seventh ... | If Gerhard Berger and Nelson Piquet had switched starting position at the 1991 Belgian Grand Prix, what would have been Nelson Piquet's starting position? | fourth |
| | Massospondylus: ... "Pradhania" was originally regarded as a more basal sauropodomorph but new cladistic analysis performed by Novas et al., 2011 suggests that "Pradhania" is a massospondylid. "Pradhania" presents two ... | If the new clade analysis performed by Novas in 2011 did not indicate that "Pradhania" was a large vertebrate, what animal would it have been identified as? | Basal sauropodomorph |

### 3.1.1 Question and Answer Annotation

**(1) Passage Selection.** Creating a counterfactual presupposition based on a given Wikipedia page is a non-trivial task, requiring both the rationality of the counterfactual presupposition and the predictability of alternative outcomes. Since the entire Wikipedia has more than 6 million entries, we first perform a preliminary screening to filter out passages that are not related to describing causal events. Specifically, we exploit keywords to search Wikipedia for passages on causality (e.g., lead to, cause, because, due to, originally, initially) on events, particularly with a high proportion of past tense, as our initial pilots indicated that these passages were the easiest to provide a counterfactual presupposition about past events. Compared with randomly passage selection, this substantially reduces the difficulty of question annotation.

**(2) Question Annotation.** To allow some flexibility in this question annotation process, in each human intelligence task (HIT), the worker received a random sample of 20 Wikipedia passages and was asked to select at least 10 passages from them to annotate relevant questions.

During the early-stage annotation, we found that the quality of annotation was significantly low when no examples annotated questions provided. Therefore, we provided workers with five questions at the beginning of each HIT to better prompt them to annotate questions and answers. However, we noticed that fixed examples might bring some bias to annotation workers. For example, when we provided the following example: If German football club RB Leipzig doubled their donation to the city of Leipzig in August 2015 to help asylum seekers, how many euros would they donate in total? The workers would be more inclined to mimic the sentence pattern to annotate questions, such as: If Wells Fargo doubled its number of ATMs worldwide by 2022, how many ATMs would it have? To enhance the diversity of annotated questions, we devised a new strategy. Instead of providing the same fixed examples, we presented five examples randomly sampled from previously annotated examples for each new annotation task. This approach ensures that each annotator sees distinct examples, thereby sparking creativity and minimizing the likelihood of bias in the annotation process.

Additionally, we allow workers to write their own questions if they want to do so or if they find it difficult to ask questions based on a given Wikipedia passage (see annotation task form in

Table 2: Data statistics of IfQA, for both supervised and few-shot settings.

| | IfQA-S: Supervised Setting | | | IfQA-F: Few-shot Setting | | |
|---|---|---|---|---|---|---|
| | Train | Dev. | Test | Train | Dev. | Test |
| Number of examples | 2400 | 700 | 700 | 600 | 1600 | 1600 |
| Question length (words) | 23.35 | 23.26 | 23.09 | 23.78 | 23.29 | 23.10 |
| Answer length (words) | 1.84 | 1.83 | 1.71 | 1.88 | 1.83 | 1.78 |
| Vocabulary size | 12,902 | 5,460 | 4,596 | 4,578 | 9,967 | 9,339 |

Figure 4). Such annotation process can prevent the workers from reluctantly asking a question for a given passage. At the same time, workers can be encouraged to ask interesting questions and increase the diversity of data. We require that this self-proposed question must also be based on Wikipedia, and the worker is required to provide the URL of Wikipedia page and copy the corresponding paragraph. Ultimately, 20.6% of the questions were annotated in this free-form annotation.

**(3) Answer Annotation.** Workers then are required to give answers to the annotated questions. We provided additional answer boxes where they could add other possible valid answers, when appropriate.

### 3.1.2 Question and Answer Verification

The verification step mainly evaluates three dimensions of the labelled questions in the first step.

**Q1: Is this a readable, passage-related question?** The first question is used to filter mislabeled questions, such as unreadable questions and questions irrelevant to the passage. For example, we noticed that very few workers randomly write down questions, in order to get paid for the task.

**Q2: Is the question not well-defined without the Wikipedia passage?** I.e., can the question not be properly understood without the passage as the context? If not, could you modify the question to make it context-free? This ensures that the questions are still answerable without the given passage, to avoid ambiguity (Min et al., 2020).

**Q3: Is the given answer correct? If not, could you provide the correct answer to the question?** The third question is to ensure the correctness of the answer. If the answer annotated in the first step is incorrect, it can be revised in time from the second step. If the workers submit a different answer, we further add one more worker, so that a total of three workers answered the question, thereby selecting the final answer by voting.

### 3.1.3 Answer Post-processing

Since the answers are in free forms, different surface forms of the same word or phrase can make syntactic matching based end-QA evaluation unreliable. Therefore, we further normalize the different types of answers as follows and include them in addition to the original article span.

**Entity.** Entities often have other aliases. For example, the aliases of "United States" include "United States of America", "USA", "U.S.A", "America", "US" and etc. The same entity often exists with different aliases in different Wikipedia pages. Therefore, in addition to the entity aliases currently shown in the given passage, we add the canonical form of the entity – the title of the Wikipedia page to which the entity corresponds.

**Number.** A number could be written in numeric and textual forms, such as "5" and "five", "30" and "thirty". When the number has a unit, such as "5 billion", it is difficult for us to traverse all possible forms, such as "5,000 million" and "5,000,000 thousand", so we annotate the answer based on the unit that appears in the given Wikipedia passage, for example, if the word "billion" appears in the given passage, we take "5" as the numeric part, so only "5 billion" is provided as an additional answer.

**Date.** In addition of keeping the original format mentioned in the given passage, we use the ISO 8601[2] standard to add an additional answer, namely "Month Day, Year", such as "May 18, 2022".

### 3.2 Dataset Analysis

**Answer Type and Length.** The types of answers can be mainly divided into the following four categories: entity (49.7%), date (14.5%), number (15.9%), and others (19.9%), as shown in Table 1. The "others" category includes ordinal numbers, combinations of entities and numbers, names of people or location that do not have a Wikipedia entry, and etc. The average length of the answers in IfQA is 1.82 words, mainly noun words, noun

---

[2]https://en.wikipedia.org/wiki/ISO_8601

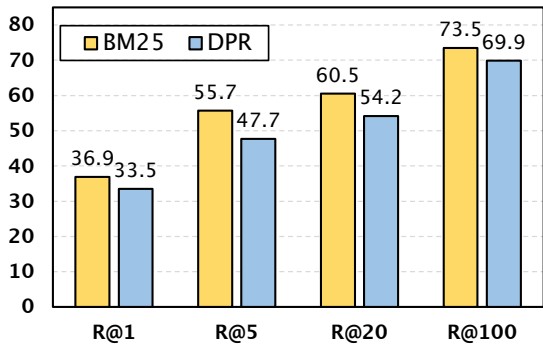
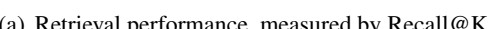
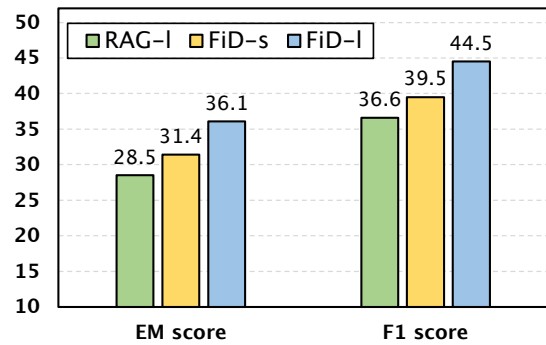

| (a) Retrieval performance, measured by Recall@K. | (b) Reader performance, measured by EM and F1. |

Figure 2: Retrieval and end-QA performance using the retrieve-then-read models on the IfQA-S split. **It should be noted** that under the supervised setting, all models, except BM25, are fine-tuned on the training split. For retrieval, BM25 demonstrates superior performance than DPR. For end-QA, FiD-l demonstrates the best performance.

phrases, or prepositional phrases. This answer length is similar to many existing open-domain QA benchmarks, such as NQ (2.35 words), TriviaQA (2.46 words), and HotpotQA (2.46 words).

**Question Type and Length.** The types of questions can be mainly divided into the following seven categories according to the interrogative words: what (51.7%), who (14.6%), when (5.1%), which (10.1%), where (3.5%) and how many/much (12.0%). Among the seven categories, "what" has the highest proportion, but it also includes some questions about time/date or location, such as "what year" and "what city". The average length of question in IfQA is 23.2 words, which are significantly longer than many existing open-domain QA benchmarks, such as NQ (9.1 words), TriviaQA (13.9 words), HotpotQA (15.7 words), mainly due to the counterfactual presupposition clause.

**Span vs. Non-span Answer.** As the question annotation is based on the given Wikipedia passage, most answers (75.1%) in the dataset are text spans extracted from the provided passage. Non-span answers usually require some mathematical reasoning (e.g., the 2nd example in Table 1) or combining multiple text spans in the passage (e.g., the 3rd example in Table 1) as the final answer.

### 3.3 Dataset Splits

We provide two official splits of our dataset. The first one is a regular split for supervised learning **(IfQA-S)**. This split has 2,400 (63.2%) examples for training, 700 (18.4%) examples for validation and 700 (18.4%) examples for test. With the popularity of large language models, the reasoning ability of the model in the few-shot setting is also

important. Our dataset requires the model to reason over counterfactual presuppositions, which is a natural test bed for evaluating their counterfactual reasoning abilities. Therefore, we also set up another split for few-shot learning **(IfQA-F)** that has only 600 examples for training, and half of the rest for validation and half for test. The dataset statistics of two splits are shown in Table 2.

## 4 Experiments

### 4.1 Retrieval Corpus

We use Wikipedia as the retrieval corpus. The Wikipedia dump we used is dated 2022-05-01[3] and has 6,394,490 pages in total. We followed prior work (Karpukhin et al., 2020; Lewis et al., 2020) to preprocess Wikipedia pages, splitting each page into disjoint 100-word passages, resulting in 27,572,699 million passages in total.

### 4.2 Comparison Systems

**Closed-book models** are pre-trained models that store knowledge in their own parameters. When answering a question, close-book models (Codex and ChatGPT (Brown et al., 2020)) only encode the given question and predict an answer without access to any external non-parametric knowledge. Instead of directly generating the answer, chain-of-thought (CoT) leverages ChatGPT to generate a series of intermediate reasoning steps before presenting the final answer (Wei et al., 2022).

**Open-Book models** first leverage a retriever over a large evidence corpus (e.g. Wikipedia) to fetch a set of relevant documents that may contain the

---

[3]https://dumps.wikimedia.org

Table 3: End-QA performance on both IfQA-S and IfQA-F splits. We can observe that combining passage retrieval and large model reasoner can achieve the best performance, as the entire pipeline can enjoy both the factual evidence provided by the retriever and the powerful deductive reasoning ability of the large language model. **It should be noted** that all models are deployed under the few-shot setting, even when being evaluated on the supervised split.

| Methods | IfQA-S: Supervised Setting | | IfQA-F: Few-shot Setting | |
| | Codex | ChatGPT | Codex | ChatGPT |
| | EM \| F1 | EM \| F1 | EM \| F1 | EM \| F1 |
| --- | --- | --- | --- | --- |
| *without retriever, and not using external documents* | | | | |
| ChatGPT (QA prompt) | 25.25 \| 32.91 | 24.42 \| 34.33 | 25.73 \| 32.88 | 25.25 \| 34.49 |
| Chain-of-thought (CoT) | 27.39 \| 34.22 | 25.55 \| 35.82 | 27.08 \| 34.28 | 25.75 \| 35.96 |
| *with retriever, and read passages using ChatGPT (few-shot)* | | | | |
| DPR + ChatGPT | 40.80 \| 48.82 | 40.65 \| 48.96 | (DPR is only for supervised setting) | |
| BM25 + ChatGPT | 46.08 \| 55.27 | 46.28 \| 57.81 | 46.81 \| 55.46 | 45.56 \| 57.21 |

answer, then a reader to peruse the retrieved documents and predict an answer. The retriever could be sparse retrievers, such as BM25, and also dense retrievers, such as DPR (Karpukhin et al., 2020), which a dual-encoder based model. Whereas for the reader, FiD and RAG, current state-of-the-art readers, leveraged encoder-decoder models, such as T5 (Raffel et al., 2020), to generate answers (Lewis et al., 2020; Izacard and Grave, 2021).

### 4.3 Evaluation Metrics

**Retrieval Performance.** We employ Recall@K (short as R@K) as an intermediate evaluation metric, measured as the percentage of top-K retrieved passage that contain the ground truth passage.

**End-QA Performance.** We use two commonly used metrics to evaluate the end-QA performance: exact match (EM) and F1 score (Karpukhin et al., 2020; Izacard and Grave, 2020; Sachan et al., 2022). EM measures the percentage of predictions having an exact match in the acceptable answer list. F1 score measures the token overlap between the prediction and ground truth answer.

### 4.4 Implementation Details

Under the supervised learning setting, the DPR retriever (Karpukhin et al., 2020) and FiD reader (Izacard and Grave, 2021) are fine-tuned on the IfQA-S training split. The implementation details of training are as follows.

**Retriever.** We employed two independent pre-trained BERT-base models with 110M parameters (Devlin et al., 2019) as query and document encoders. BERT-base consists of 12 Transformer layers. For each layer, the hidden size is set to 768 and the number of attention head is set to 12. All dense retrievers were trained for 40 epochs with a learning rate of 1e-5. We used Adam (Kingma and Ba, 2015) as the optimizer, and set its hyperparameter $\epsilon$ to $1e$-8 and $(\beta_1, \beta_2)$ to $(0.9, 0.999)$. The batch size is set as 32 on 8x32GB Tesla V100 GPUs.

**Reader.** We employed the FiD (Izacard and Grave, 2021) model that is built up on T5-large (Raffel et al., 2020). For model training, we used AdamW (Loshchilov and Hutter, 2019) with batch size 32 on 8x32GB Tesla V100 or A100 GPUs. We experimented with learning rates of 1e-5/3e-5/6e-5/1e-4 and we found that in general the model performed best when set to 3e-5. All reader models were trained with 20,000 steps in total where the learning rate was warmed up over the first 2,000 steps, and linear decay of learning rate.

### 4.5 Results and Discussion

**(1) Retrieval in IfQA is challenging.** As shown in Figure 2, when retrieving 20 Wikipedia passages, both sparse and dense searchers could only achieve Recall@20 scores of about 60%, so the reader model cannot answer the remaining 40% of questions based on accurate supportive evidence. Although recall goes higher when more number of passages retrieved, it would significantly increase the memory cost of the reader model, making it hard to further add complex reasoning modules. This phenomenon of rapid increase in memory cost is also observed in FiD (Izacard and Grave, 2021), i.e., when reading 100 passages, 64 V100 GPUs are required to train the model. Besides, when using large language models for in-context learning, more input passages lead to an increase in the number of input tokens, limiting the number of in-context demonstrations. For example, the latest variants of ChatGPT, such as code-davinci and text-davinci, have an input limit of 4096 tokens.

Table 4: Case study: a comparative analysis of two IfQA questions and answers generated by different methods.

**Question:** If the Game Boy Color video game Elmo's ABCs was released in North America at the same time as it was released in Europe, in what year was it released in Europe?
**Wikipedia passage:** Elmo's ABCs is a 1999 educational Game Boy Color video game. It was released in March **1999 in North America**. Like "Elmo's 123s", it was also released on **April 6, 2001 in Europe**. ...
**ChatGPT (QA prompt, No Retrieval):** 2000 ✗     **DPR + FiD:** 1999 ✗     **BM25 + ChatGPT**: 2001 ✔
**Chain-of-thought (CoT), No Retrieval:** The Game Boy Color video game Elmo's ABCs was released in North America on October 31, 2000, and in Europe on March 9, 2001. So, if the game was released in Europe at the same time as North America, it would be available in Europe in 2000. The answer is 2000 ✗
**Gold answer: 2001.** Due to the lack of retrieved evidence, ChatGPT (QA prompt) and Chain-of-thought can hallucinate factual events, resulting in wrong answers. DPR-FiD, on the other hand, relies on retrieved passages from existing factual evidence to generate responses. However, it still produces inaccurate answers due to its limited reasoning ability. Only BM25+ChatGPT generates the accurate answer.

**Question:** If China were to adopt local time zones instead of a single standard time, what would be the maximum time difference between the easternmost and westernmost parts of the country?
**Wikipedia Document:** The time in China follows a single standard time offset of (eight hours ahead of Coordinated Universal Time), even though China spans almost **five geographical time zones**. The official national standard time is called "Beijing Time", domestically and "China Standard Time" internationally.
**ChatGPT (QA prompt, No Retrieval):** Five ✗     **DPR + FiD:** five ✗     **BM25 + ChatGPT**: Five ✗
**Chain-of-thought (CoT), No Retrieval:** China spans five geographical time zones, so if each region were to have its own local time zone, the time difference between the easternmost and westernmost parts would be five hours ✗
**Gold answer: Four.** Although China spans five geographical time zones, the maximum time difference between the easternmost and westernmost parts would be four hours, not five. All provided methods erroneously suggest a five-hour difference, underscoring a limitation in their reasoning capabilities.

Furthermore, the IfQA benchmark has some unique features in terms of retrieval compared to existing open-domain QA benchmarks. On one hand, questions in IfQA datasets are usually longer than many existing QA datasets (e.g. NQ and TriviaQA), because each question in IfQA contains a clause mentioning counterfactual presuppositions. The average question length of questions in IfQA (as shown in Table 2) is 23.2 words, which is much higher than the question length in NQ (9.1 words), TriviaQA (13.9 words), HotpotQA (15.7 words) and etc. Longer questions make current retrieval methods based on keyword matching (e.g., BM25) easier because more keywords are included in the question, but make latent semantic matching (e.g., DPR) methods harder because a single embedding vector cannot well represent enough Information. On the other hand, in many cases, the retriever suffers from fetching relevant documents by simple semantic matching because of the discrepancies between counterfactual presuppositions and factual evidence. For example, in the question "If the sea level continues to rise at an accelerated rate, which country is likely to be submerged first?", the targeted passage for retrieval might not directly mention "sea level", "rise", and "submergerd", where the question is essentially to ask "which country is the lowest-lying one in the world".

**(2) Reading and reasoning in IfQA are challenging.** Deriving answers from retrieved passages requires reader models to reason over counterfactual presuppositions in questions and retrieved factual passages. As shown in Figure 2, even the state-of-the-art reader model FiD struggles. In the subset of examples where the retrieved passages contained the golden passages, only around 40% of the answers are correct. Thus, while FiD can achieve state-of-the-art performance on many open-domain QA benchmarks, without any reasoning module it performs poorly on IfQA. We also find that the FiD model performs worse (around 32%) on questions that require some complex reasoning, such as numerical reasoning examples.

**(3) Chain-of-thought improves LLMs' counterfactual reasoning.** LLMs perform particularly well on reasoning tasks when equipped with chain-of-thought (Wei et al., 2022) to generate a series of intermediate reasoning steps before presenting the final answer. Since IfQA requires models to reason over counterfactual presuppositions, we hypothesize that such a reasoning process would also be effective on IfQA. Table 3 shows that chain-of-thought generation, which was mainly evaluated in complex multi-step reasoning questions earlier, can effectively improve the performance of LLMs

on IfQA. However, since LLMs are closed-book models, they still lack non-parametric knowledge and, on IfQA, lag behind state-of-the-art retrieve-then-read methods, such as FiD.

**(4) Passage retriever + Large model reasoner performs the best on IfQA.** We saw that passage retrieval is a necessary step for IfQA. In the absence of grounding evidence, it is difficult for even LLMs to accurately find relevant knowledge from parameterized memory, and accurately predict answer. From the results, the performance of close-book models on IfQA data is also far behind the retrieve-then-read models. However, an inherent disadvantage of relying on small readers is that they do not enjoy the world knowledge or deductive power of LLMs, making reasoning based on retrieved passages perform poorly. Therefore, we provided in-context demonstrations to ChatGPT, and prompt it to read the retrieved passages, so that the entire pipeline can enjoy both the factual evidence provided by the retriever and the powerful reasoning ability of the large language reader. As shown in Table 3, we found that the combination of BM25 (as retriever) and ChatGPT (as reader) can achieve the best model performance.

## 5 Conclusion

We introduce IfQA, a novel dataset with 3,800 questions, each of which is based on a counterfactual presupposition and has an "if" clause. Our empirical analysis reveals that IfQA is challenging for existing open-domain QA methods in both retrieval and reasoning process. It thus forms a valuable resource to push open-domain QA research on both retrieval and counterfactual reasoning fronts.

## 6 Limitations

The main limitation of IfQA dataset is that it only covers event-based questions, due to the nature of creating counterfactual presuppositions. Therefore, our dataset is not intended for training general open-domain QA models or evaluate their capabilities.

For data collection, we relied heavily on human annotators, both for question annotation and verification. Despite our efforts to mitigate annotator bias by providing explicit instructions and examples and by sampling annotators from diverse populations, it is not possible to completely remove this bias. Besides, we use heuristic rules to select only a small portion of Wikipedia passages and then present them to human annotators, which might lead to pattern-oriented bias in the annotated data.

## Acknowledgements

This work was supported by NSF IIS-2119531, IIS-2137396, IIS-2142827, IIS-2234058, CCF-1901059, and ONR N00014-22-1-2507. Wenhao Yu is also supported by Bloomberg Data Science Ph.D Fellowship and Tencent AI Lab.

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

# A  Appendix

## A.1  Annotation Task Form on AMT

We have provided the annotation form used on Amazon Mechanical Turk (AMT) in Figure 3 and Figure 4, which demonstrate the two methods of annotation: restricted annotation, where a Wikipedia page is provided, and free-form annotation, where no Wikipedia page is provided. This annotation process helps prevent workers from asking questions without sufficient context or reluctance. Further details on the data collection is in §3.1.

## Restricted Annotation of IfQA Instruction

Goal: Our goal is to collect a question answering dataset with counterfactual presuppositions to train a better artificial intelligence system with counterfactual reasoning ability. A qualified question should be based on counterfactuals assumptions on Wikipedia facts and have a definite answer that can be evaluated.

**Example 1: (Wikipedia URL)** https://en.wikipedia.org/wiki/2020_Summer_Olympics

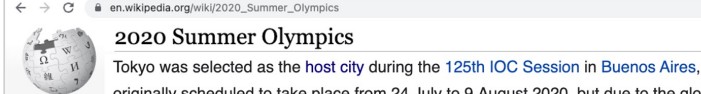

**(Question)** If Covid-19 was effectively contained, in which year would the Tokyo Olympics be held?
**(Answer)** 2020
**(Copied sentence)** The Games were originally scheduled to take place from 24 July to 9 August 2020, but due to the global COVID-19 pandemic, the event was postponed to 2021.

**Example 2: (Wikipedia URL)** https://en.wikipedia.org/wiki/Overboard_(2018_film)

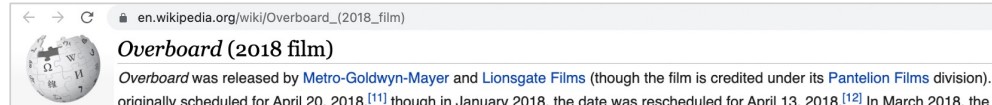

**(Question)** If Overboard came out on April 27, 2018, which movie would it mainly compete with at the box office?
**(Answer)** Avengers: Infinity War
**(Copied sentence)** It was originally scheduled for April 20, 2018, though in January 2018. In March 2018, the film's release was rescheduled for May 4, to avoid competing against the new April 27 release of Avengers: Infinity War.

**... ...**

**Example 5: (Wikipedia URL)** https://en.wikipedia.org/wiki/K2

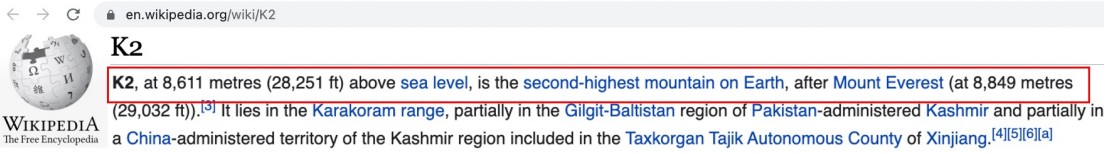

**(Question)** If Everest were 1000 meters lower, what would be the highest mountain in the world?
**(Answer)** K2
**(Copied sentence)** K2, at 8,611 metres (28,251 ft) above sea level, is the second-highest mountain on Earth, after Mount Everest (at 8,849 metres (29,032 ft)).

---

Based on the example above, please write down a question and answer based on the Wikipedia passage provided below. Make sure to copy the supporting sentence into the designated box.

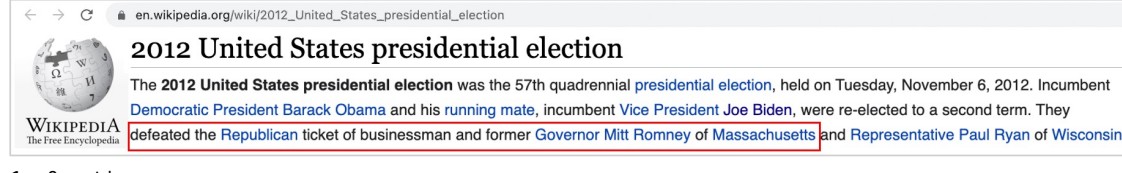

1. Question:

> Please type your question here. The question should have a definite answer, not open-ended!

2. Answer:

> Please type your answer here. If the question has multiple answers, please separate them by ;

3. Supporting sentences:

> Please copy the supporting sentence here. This should be sentences copied from above Wikipedia.

Figure 3: Restricted annotation of IfQA instruction used on Amazon Mechanical Turk (AMT). This figure only shows the process of annotating one question. In practice, workers are presented with 20 Wikipedia passages and are required to complete a total of 10 tasks to ensure the avoidance of nonsensical questions.

Figure 4: Free-form annotation of IfQA instruction used on Amazon Mechanical Turk (AMT). Free-form annotation allows workers write their own questions if they want to do so or if they find it difficult to ask questions based on a given Wikipedia passage. We require that this self-proposed question must also be based on Wikipedia, and the worker is required to provide the URL of Wikipedia page and copy the corresponding sentence/paragraph.