# OpenReview forum: "IfQA: A Dataset for Open-domain Question Answering under Counterfactual Presuppositions"
_EMNLP/2023/Conference — EMNLP 2023 Main_

### Official Review · Reviewer_sGbp · 2023-08-04

**Soundness:** 3

**Excitement:**

3: Ambivalent: It has merits (e.g., it reports state-of-the-art results, the idea is nice), but there are key weaknesses (e.g., it describes incremental work), and it can significantly benefit from another round of revision. However, I won't object to accepting it if my co-reviewers champion it.

**Paper Topic And Main Contributions:**

The paper seems to delve into the realm of question answering with a specific focus on counterfactual reasoning. The goal, as outlined, is to create a dataset for question answering that is based on counterfactual presuppositions. This is aimed at training AI systems to have a better counterfactual reasoning ability. The paper provides examples of questions that are based on counterfactual assumptions on Wikipedia facts. The paper also touches upon the annotation process, using platforms like Amazon Mechanical Turk, to ensure the quality of the questions.

**Questions For The Authors:**

How does this work compare with other counterfactual reasoning approaches in the literature?

**Reasons To Accept:**

1. The paper addresses a novel and challenging area in NLP, which is counterfactual reasoning in question answering.
2. The approach of creating a dataset based on counterfactual presuppositions can be a valuable resource for the NLP community.
3. The inclusion of a detailed annotation process ensures the quality and reliability of the dataset.

**Reasons To Reject:**

1. The paper seems to rely heavily on Wikipedia as a source, which might limit its applicability in more diverse contexts.
2. The evaluation metrics and results are not clearly mentioned in the extracted content, making it hard to gauge the effectiveness of the proposed approach.

**Reproducibility:**

3: Could reproduce the results with some difficulty. The settings of parameters are underspecified or subjectively determined; the training/evaluation data are not widely available.

**Reviewer Confidence:**

3: Pretty sure, but there's a chance I missed something. Although I have a good feel for this area in general, I did not carefully check the paper's details, e.g., the math, experimental design, or novelty.

---

> ### Author Rebuttal · Authors · 2023-08-27
>
> Thank you for your feedback. We are very encouraged that you found our dataset novel and valuable to the community. We address your concerns as follows.
>
> > 1.The paper seems to rely heavily on Wikipedia as a source, which might limit its applicability in more diverse contexts.
>
> -- While we agree that using other sources in addition to Wikipedia would make the benchmark even stronger, we believe Wikipedia already provides diverse enough topics and extensive coverage to build a meaningful dataset to develop and evaluate novel models. This is akin to numerous existing open-domain QA benchmarks which also use Wikipedia as the underlying knowledge source. Secondly, the foundational challenge introduced by our IfQA dataset can be extrapolated to other domains as well, namely evaluating a model's ability to identify the right information to retrieve and reason about an imagined situation that may even go against the facts built into its parameters.
>
> > 2.The evaluation metrics and results are not clearly mentioned in the extracted content, making it hard to gauge the effectiveness of the proposed approach.
>
> -- We note that we didn’t fully understand what the reviewer meant by “extracted content”. We will try to answer the question as best as we can here, but are happy to expand further.
>
> The evaluation metrics are listed in section 4.3 and experiential results in section 4.4. In particular, the retrieval performance is evaluated as recall@k, the percentage of top-K retrieved passages that contain the ground truth passage. The end-QA Performance is evaluated by standard exact match and F1 score metrics, which measure the token overlap between the prediction and the ground truth answer.
>
> > 3.How does this work compare with other counterfactual reasoning approaches in the literature?
>
> -- Our work primarily introduces a novel counterfactual QA dataset, which is the first dataset of open-domain QA focused on counterfactual reasoning. We compare our paper with prior work in the area of counterfactual reasoning in section 2.2. We will clarify that prior work is either about models (whereas ours is about a new benchmark), or about counterfactual datasets in the reading comprehension setting (while ours is in the open-domain setting).
> Regarding counterfactual reasoning methods, traditional approaches in existing literature are not directly applicable due to the unique challenges presented by IfQA. To navigate this open-domain characteristic effectively, we mainly experimented with retrieve-then-read pipelines, since they are SoTA for existing benchmarks. However, they are not geared to tackle such challenging reasoning tasks. IfQA requires models to not only pinpoint the relevant information to retrieve but also to cogitate over counterfactual scenarios, some of which might challenge or even contradict the inherent facts within their training data.

---

### Official Review · Reviewer_hauT · 2023-08-05

**Typos Grammar Style And Presentation Improvements:** Typos on line 42-43 and line 501.
**Soundness:** 4

**Excitement:**

4: Strong: This paper deepens the understanding of some phenomenon or lowers the barriers to an existing research direction.

**Paper Topic And Main Contributions:**

This paper presents a dataset of counterfactual questions and answers (challenging for existing retrieval/open-book and closed-book models). Counterfactuals are particularly challenging for LLMs because they rely on a combination of world-sense/reasoning and retrieval. The authors also establish an initial set of performance baselines with state of-the-art close-book and open-book models (e.g., chain-of-thought GPT-3, RAG, FiD, etc.). This paper demonstrates that the set of counterfactuals presented by IfQA remains a significant challenge for existing models, finding that there is a large gap between human performance and the best approach (passage retrieval + large model reasoner).

**Reasons To Accept:**

Counterfactual reasoning is an important strength of human intelligence and to achieve more robust question answering models, it is important to have a publically available dataset/set of baseline metrics. This paper presents clear and concise insights into how the dataset was built as well as an in-depth analysis into the dataset and how a broad set of state-of-the-art models perform on the given task. The discussion is thorough and insightful and most design choices are motived/justified well.

**Reasons To Reject:**

Aside from common concerns in most crowdsourced dataset papers (e.g., bias in views, human error due to limited validation resources, etc.), the weaknesses of this paper (or other suggestions) can be summarized as:
1. There should be some analysis on similarity of questions (maybe in Appendix) because the 5 randomly sampled examples may still have an impact on annotation quality. For example, replace named entities with their respective category and use some similarity metric (e.g., word embs, WMD) to cluster to see how diverse annotations actually are. Generally, they should be evenly spaced for diverse questions.
2. There is a lot of overlap in counterfactual questions and in other tasks such as common-sense QA and mathematics word question answering. As mentioned in the paper, the keyword filter on Wikipedia seems to have made the dataset heavily skewed towards event-related questions. It would be interesting to see these aforementioned domains being included in the dataset. For example, many math questions present counterfactuals which can be expanded to things like corporate decision making. If this is not the goal, it may be more apt to make the dataset more specific - "Event-based counterfactual open-domain QA".


**Reproducibility:**

N/A: Doesn't apply, since the paper does not include empirical results.

**Reviewer Confidence:**

4: Quite sure. I tried to check the important points carefully. It's unlikely, though conceivable, that I missed something that should affect my ratings.

---

> ### Author Rebuttal · Authors · 2023-08-27
>
> Thank you for your positive feedback. We are very encouraged that you found our dataset novel and valuable to the community. We address your concerns as follows.
>
> > 1.There should be some analysis on similarity of questions because the 5 randomly sampled examples may still have an impact on annotation quality. For example, replace named entities with their respective category and use some similarity metric (e.g., word embs, WMD) to cluster to see how diverse annotations actually are. Generally, they should be evenly spaced for diverse questions.
>
> -- Following your recommendation, we organized the training data into two sets of clusters: one set with 5 clusters and another with 10. We observed that the distribution of instances across these clusters was relatively even. Below are the details:
>
> | #clusters | mean | std |  #1 |  #2 |  #3 |  #4 |  #5 |  #6 |  #7 |  #8 |  #9 | #10 |
> |:---------:|:----:|:---:|:---:|:---:|:---:|:---:|:---:|:---:|:---:|:---:|:---:|:---:|
> |     5     |  480 |  88 | 541 | 468 | 405 | 542 | 447 |     |     |     |     |     |
> |     10    |  240 |  47 | 257 | 209 | 206 | 240 | 250 | 210 | 251 | 254 | 255 | 271 |
>
>
> > 2.There is a lot of overlap in counterfactual questions and in other tasks such as common-sense QA and mathematics word question answering. As mentioned in the paper, the keyword filter on Wikipedia seems to have made the dataset heavily skewed towards event-related questions. It would be interesting to see these aforementioned domains being included in the dataset. For example, many math questions present counterfactuals which can be expanded to things like corporate decision making. If this is not the goal, it may be more apt to make the dataset more specific - "Event-based counterfactual open-domain QA".
>
>
> -- While a significant portion of the IfQA dataset comprises event-based questions, owing to their relevance to real-world situations and potential alternative outcomes, the scope of IfQA extends beyond events. As highlighted by Figure 1 and Table 1, our dataset also incorporates questions that necessitate both commonsense reasoning and mathematical proficiency.
>
> We note that an important distinguishing aspect of IfQA is that it’s based on verifiable (counterfactual) facts in an open-domain setting. E.g., for the question “who would have become the president of the USA if Obama lost in 2008?”, there is a unique factual answer (John McCain), deriving which requires external IR and reasoning. This is generally not the case for commonsense QA or math word problems, even if they are phrase as “if” questions (e.g., “if John had 3 apples and gave 1 to Mary, …”). Answering some counterfactual questions might call upon commonsense or math skills, but the challenge generally goes further – models must identify the right information to retrieve from an external source and reason in a way that might even go against facts built into their parameters.
>
> Thus, while there are some intersections between counterfactual QA and commonsense or mathematical QA, the main challenge of IfQA is orthogonal to them.

---

### Official Review · Reviewer_8qae · 2023-08-07

**Soundness:** 4

**Excitement:**

4: Strong: This paper deepens the understanding of some phenomenon or lowers the barriers to an existing research direction.

**Paper Topic And Main Contributions:**

The paper provides a new dataset in the domain of QA, but based on counterfactual reasoning. The author(s) show that current existing methods are not geared to tackle such reasoning, and thus perform quite low.

**Reasons To Accept:**

1. The paper presents a very important aspect of reasoning that human tends to do in certain scenarios. So, this type of "thinking" is a crucial step for AGI development. I believe this dataset would spark interest in this direction and lead to fruitful research contribution.

**Reasons To Reject:**

1. The author(s) could have provided a direction of thought or discussion as to their perspective, how current techniques (or newer ones) might be modeled to tackle such counterfactual reasoning scenarios.

**Reproducibility:**

4: Could mostly reproduce the results, but there may be some variation because of sample variance or minor variations in their interpretation of the protocol or method.

**Reviewer Confidence:**

3: Pretty sure, but there's a chance I missed something. Although I have a good feel for this area in general, I did not carefully check the paper's details, e.g., the math, experimental design, or novelty.

---

> ### Author Rebuttal · Authors · 2023-08-27
>
> Thank you for the constructive feedback. We concur that providing a direction of thought for future techniques is valuable.  Here are some promising directions, which we would be happy to include in the paper:
>
>
> > 1.The author(s) could have provided a direction of thought or discussion as to their perspective, how current techniques (or newer ones) might be modeled to tackle such counterfactual reasoning scenarios.
>
>
> **-- Overcoming Retrieval challenge:** (1) Questions in the IfQA dataset are typically longer, owing to the inclusion of counterfactual presuppositions. This causes challenges for current IR models, such as DPR, which tend to utilize a single embedding vector to represent comprehensive information. One promising way to get around this limitation is to consider multi-vector representations that capture different nuances in the question, and explicitly perform IR for both the original fact as well as the presupposition.
>
> (2) The inherent discrepancies between counterfactual presuppositions and factual evidence frequently cause the retriever to falter in obtaining relevant documents when relying solely on semantic matching. Incorporating clear instructions for retrieval might help bridge the gap between the counterfactual and factual worlds.
>
> **-- Overcoming Reasoning Challenge:** For counterfactual questions, models must perform complex reasoning on potential situations that diverge from the factual data built into their parameters. This may be achieved via a multi-stage reasoning mechanism, wherein the initial stages deal with understanding and extracting the counterfactual premise, and the subsequent stages handle the implications of that premise on the original reasoning.
>
> **-- Adaptive Chain-of-Thought Reasoning:** We showed the chain-of-thought reasoning approach offers a potential avenue for counterfactual reasoning. However, it still struggles with many IfQA questions.One way to improve it could be to employ adaptive reasoning chains that dynamically adjust based on the nature of the counterfactual premise. This could involve iterative questioning or even self-dialogue approaches where the model questions its assumptions and iteratively refines its reasoning process.
>
> We will enhance the paper by incorporating a more detailed discussion on these fronts, elaborating on both the current limitations of models on IfQA and potential pathways to improve performance. Once again, thank you for the suggestion!

---

### Meta-Review · Area_Chair_mKUv · 2023-09-07

**Recommendation:** 5

**Metareview:**

This paper presents a new dataset for open-domain question answering where the questions contain counterfactual/hypothetical situations. The model must retrieve passages based on these counterfactual situations to correctly answer the question. The reviewers found that the proposed dataset targets an important ability for ODQA systems and demonstrates that existing state-of-the-art systems perform poorly on the task. They also found that the paper presents sufficient detail and provides interesting insights about the dataset and models.

---

### Decision · Program_Chairs · 2023-10-07

**Decision:**

Accept-Main

**Comment:**

This paper presents a new dataset for open-domain question answering where the questions contain counterfactual/hypothetical situations. The model must retrieve passages based on these counterfactual situations to correctly answer the question. The reviewers found that the proposed dataset targets an important ability for ODQA systems and demonstrates that existing state-of-the-art systems perform poorly on the task. They also found that the paper presents sufficient detail and provides interesting insights about the dataset and models.